# Evaluation of Perceptual Interactions between Ester Aroma Components in Langjiu by GC-MS, GC-O, Sensory Analysis, and Vector Model

**DOI:** 10.3390/foods9020183

**Published:** 2020-02-13

**Authors:** Yunwei Niu, Ying Liu, Zuobing Xiao

**Affiliations:** 1School of Perfume and Aroma Technology, Shanghai Institute of Technology, No.100 Haiquan Road, Shanghai 201418, China; nyw@sit.edu.cn (Y.N.); 176071213@mail.sit.edu.cn (Y.L.); 2Beijing Advanced Innovation Center for Food Nutrition and Human Health, No. 11/33, Fucheng Road, Haidian District, Beijing 100048, China

**Keywords:** Langjiu, ester compounds, perceptual interactions, Feller’s additive model, OAV, vector model

## Abstract

The volatile compounds of three Langjiu (“Honghualangshi, HHL”, “Zhenpinlang, ZPL”, and “Langpailangjiu, LPLJ”) were studied by gas chromatography-olfactometry (GC-O) and gas chromatography-mass spectrometry (GC-MS). The results showed that a total of 31, 30, and 30 ester compounds making a contribution to aroma were present in the HHL, ZPL, and LPLJ samples, respectively. From these esters, 16 compounds were identified as important odour substances, and their odour activity values (OAVs) were greater than 1. The key ester components were selected as: ethyl acetate, ethyl 2-methylbutyrate, ethyl 3-methyl butyrate, ethyl hexanoate, and ethyl phenylacetate by aroma extract dilution analysis (AEDA), odour activity value (OAV), and omission testing. Five esters were studied for perceptual interactions while using Feller’s additive model, OAV, and a vector model. Among these mixtures, they all have an enhancing or synergistic effect. Among these mixtures, one mixture presented an additive effect and nine mixtures showed a synergistic effect.

## 1. Introduction

Chinese liquor, which is a popular alcoholic beverage in China, is composed of ethanol, water, and trace components. Ethanol and water account for 98% to 99% of the total mass of liquor, and the remaining 1% to 2% is composed of trace compounds. It is this 1% to 2% (the trace components) that can determine the flavour and style of the liquor. Chinese liquor consists of three aroma types according to the diversity of aromas: a sauce-aroma, strong-aroma, and light-aroma. Langjiu is sauce-aroma type of liquor. They are made from sorghum, wheat, etc., and they are fermented, distilled, aged, and blended by traditional solid-state methods [1]. The content of esters in different flavoured liquors varies, and the general ester components account for 35% to 70% of the total aroma components [2].

In recent years, research into the aroma of liquor has been reported: Cheng et al. [3] used a headspace-solid phase microextraction (HS-SPME)-mass spectrometry (MS) technique, partial least squares discriminant analysis (PLS-DA), and stepwise linear discriminant analysis (SLDA) methods to determine 131 Chinese liquor samples. Finally, 32 characteristic ions were selected, and 32 ions were then input; eight groups of ions of different geographical origins were used as outputs to establish an artificial neural network (ANN) recognition model. Liu et al. [4] introduced the direct analysis of Langjiu and its serial products by WH-3 glass chromatographic column. The application of such methods could rapidly detect more than 20 kinds of main flavour compositions, including acids, esters, alcohols (especially acid substance such as lactic acid, etc.) without the pre-treatment of liquor samples. Wei, L. and Zhang, L. [5] used methylene chloride as an extracting agent to detect the main flavouring ingredients of Langjiu by gas chromatography/mass spectrometry (GC/MS). Through the optimisation of GC/MS conditions, satisfactory separation chromatograms were obtained, each flavour component was identified based on the separation chromatograms in combination with the NIST MS library, and their relative content was then measured by using chromatographic peak area normalisation. As a result, there were 40 kinds of trace components that were identified in this study. Among them, the amounts of ethyl hexanoate, hexanoate acid, ethyl lactate, acetic acid, and butyric acid were relatively high.

Esters are not only important aroma components in liquor, but they also interact with each other. Niu et al. [6,7] studied perceptual interactions between esters. In light aroma-type liquor, triangulation test experiments were conducted to find the odour thresholds of 18 esters and 35 binary mixtures. Among them, 31 binary mixtures had synergistic effects or additive effects. Sensory analysis indicated that different concentrations of ethyl phenylacetate had a masking effect of fruit note, while the addition of phenylethyl phenylacetate at low and high concentrations promoted a floral note. The sweet note was enhanced when phenylethyl acetate was added near the threshold. In Chinese cherry wines, the addition of esters can reduce the olfactory threshold of aromatic recombination: different added amounts have significant effects on fruity, floral, sweet, and fermentation aroma intensity. In addition, Gao et al. [8] revealed the importance of the entire group of esters in liquor through omission testing. However, previous work has not elucidated the interaction between aroma compounds in Langjiu. Therefore, the main task of the present work was to study the interaction between important esters, according to three sensory analysis methods, to provide theoretical support for the analysis of the aroma of Langjiu, and provide guidance for improving the quality of Langjiu.

The primary aims of this study were: (a) to identify the ester components of three different Langjiu, qualitatively quantify them by GC-MS, gas chromatography-olfactometry (GC-O); (b) to select those important ester compounds in Langjiu by aroma extract dilution analysis (AEDA), odour activity value (OAV), and omission test; and, (c) to study the perceptual interactions between ester compounds by the Feller’s additive model, OAV, and a vector model. 

## 2. Material and Methods

### 2.1. Samples

Liquor samples were commercially obtained. The three types of Langjiu were investigated: Honghualang (HHL, 500 mL, 53% ethanol by volume, from Sichuan Langjiu Co., Ltd, Sichuan Province, China), Zhenpinlang (ZPL, 500 mL, 53% ethanol by volume, from Sichuan Langjiu Co., Ltd, Sichuan Province, China), and Langpailangjiu (LPLJ, 500 mL, 53% ethanol by volume, from Sichuan Langjiu Co., Ltd, Sichuan Province, China). All of the specimens were stored at 4 °C for further analysis.

### 2.2. Chemicals

Ethyl acetate, ethyl 2-methylpropionate, propyl acetate, ethyl propionate, ethyl butyrate, isobutyl acetate, ethyl 2-methylbutyrate, ethyl 3-methyl butyrate, isoamyl acetate, ethyl valerate, butyl butyrate, ethyl hexanoate, isoamyl butyrate, propyl hexanoate, ethyl heptanoate, ethyl lactate, 3-methylbutyl hexanoate, propyl octanoate, ethyl nonanoate, hexyl hexanoate, ethyl 2-furoate, ethyl caprate, ethyl benzoate, ethyl phenylacetate, ethyl laurate, ethyl 3-phenylpropanoate, ethyl tetradecanoate, ethyl pentadecanoate, ethyl palmitate, ethyl oleate, and linoleic acid ethyl ester were obtained from Sigma-Aldrich (Shanghai, China). The internal standard (IS) was 2-octanol (Sigma-Aldrich, Shanghai, China). The linear retention index (RI) was determined with a C7–C30 *n*-alkane mixture (Sigma-Aldrich, Shanghai, China). All of the reagents used were of analytical grade with a purity of at least 97%, and most with a purity exceeding 99%. A Milli-Q purification system provided pure water (Millipore, Bedford, MA, USA). Sodium chloride (analytical grade) and absolute ethanol (analytical grade) were obtained from Sino-pharm Chemical Reagent Co., Ltd (Shanghai, China).

### 2.3. Extraction of Volatile Compounds of Langjiu by Headspace Solid-phase Microextraction (HS-SPME)

The volatile compounds were extracted by HS-SPME, as follows: three liquor samples were diluted with deionised water to a 10% ethanol concentration. We added 8 mL liquor sample, 1.5 g sodium chloride, and 50 µL internal standard (2-octanol, 400 mg/L) to the 15 mL headspace bottle that had a PTFE−silicone septum, and then put the headspace bottle in a constant temperature water bath at 50 °C. A 50/30 µm divinylbenzene/carboxyl/polydimethylsiloxane (DVB/CAR/PDMS) fiber was exposed in the headspace without stirring for 50 min., and then desorbed into the injection port of the gas chromatograph for 5 min. At the end of each analysis, the fiber was inserted into a thermal heater at 250 °C for 20 min. to ensure that there was no residue. Each liquor sample went through the same process, as described above.

### 2.4. Identification by GC-MS and GC-O

GC-MS analysis was conducted on a 7890 gas chromatograph (GC) coupled to a 5973C mass (MS) (Agilent Technologies, Santa Clara, CA, USA). GC-O analysis used an Agilent 7890A gas chromatograph (GC), which was equipped with a Gerstel ODP2 detector (Mülheim a der Ruhr, Germany).

#### 2.4.1. GC-MS Analysis

The liquor samples were analysed while using two types of columns: an HP-Innowax column (60 m × 0.25 mm × 0.2 µm; Agilent) and a DB-5 column (60 m × 0.25 mm × 0.25 µm; Agilent). Using helium (purity 99.999%) as a carrier gas, the flow rate was 1 mL/min. The quadrupole mass filter has a temperature of 150 °C and a transfer line temperature of 250 °C [9]. The oven temperature was set to 40 °C (6 min), ramped at 3 °C/min. to 100 °C, and then ramped at 5 °C/min. to 230 °C (20 min). Mass spectrometry conditions were as follows: electron ionisation (EI) mode at 70 eV (ion source temperature 230 °C) was used and the scan range was m/z 30–450. Volatile components were identified by comparing the retention index (RI), molecular weights, and mass fragmentation patterns in the database (Wiley 7n.L Database, NIST Database) to authentic standards.

#### 2.4.2. GC-O Analysis

After the liquor sample enters the gas chromatograph, it was separated by the chromatographic column and then flowed to the detector and olfactory orifice at 1:1, respectively. The chromatographic columns were an HP-Innowax (60 m × 0.25 mm × 0.25 µm; Agilent) and a DB-5 (60 m × 0.25 mm × 0.25 µm; Agilent). Using hydrogen as the carrier gas, the flow rate was 2mL/min. The oven temperature was set to 40 °C (6 min), ramped at 3 °C/min. to 100 °C, and ramped at 5 °C/min. to 230 °C (20 min). The injector temperature was set to 250 °C and the FID temperature was set to 280 °C. In addition, the moist air entered the sniffing port at a flow rate of 50 mL/min. to expel residual aroma compounds from the sniffing port [10]. Each aroma compound was determined by comparing the RI, the odour, and mass spectra of the standard products. The FD coefficient represents the maximum dilution coefficient of each compound (Table 1). All of the trials were carried out on each liquor sample three times.

#### 2.4.3. Aroma Extract Dilution Analysis

For AEDA, the liquor samples were diluted with deionised water, and the sample with an ethanol content of 10 (*v/v*) was obtained. The sample was gradually diluted with 10% ethanol and water (1:1) until reaching 1:1024. Each dilution was submitted to GC-O analysis under the same GC conditions that are described above until no odorant was detected. The flavour dilution (FD) factor of each compound represented the maximum dilution at which the odorant could be perceived. The identification of each aroma compound was conducted by comparing their odours, RI, and mass spectra with those of pure standards. All of the trials were carried out on each liquor sample three times.

### 2.5. Quantitative Analysis

Thirty-one aroma compounds were quantified from the calibration curves. Using the prepared model liquor sample, the standard substance of appropriate concentration was added, and then diluted into six concentration gradients in turn, each concentration gradient point was extracted and then analysed three times, followed by the addition of internal standard solution (2-octane, 50 µL, 400 mg/L) to establish the calibration curves of the aroma substance for determining the aroma. It was used to determine the actual concentration of aroma substances in each liquor sample. The extraction conditions for solid phase microextraction (SPME) were the same as those of Langjiu. Table 2 lists the coefficients of the calibration curves, where *y* represents the peak area ratio (peak area of volatile standard/peak area of internal standard) and *x* denotes the concentration ratio (concentration of volatile standard/concentration of internal standard).

### 2.6. Sensory Analyses

#### 2.6.1. General Conditions

Sensory analysis was performed on behalf of Martin and Revel [11] (1999). The 10 mL sample was placed in a brown glass bottle, randomly numbered while using three digits, and then evaluated in different compartments at room temperature (20 °C).

#### 2.6.2. Sensory Panels

The assessors were grouped into sensory panel A (10 males and 10 females) and sensory panel B (two males and two females). Sensory panel A participated in the determination of threshold and model establishment, and sensory panel B participated in the determination of the dilution factor by GC-O. Sensory panels consisted of 24 volunteers (12 males and 12 females, aged between 23 and 29 years). The volunteers were selected from 40 candidates based on their performance in several olfactory tests. They suffered no problems, such as olfactory allergies. All of the volunteers were from the School of Perfume and Aroma Technology, Shanghai Institute of Technology. They attended meetings twice a week for four weeks.

#### 2.6.3. Omission Analysis

Triangular tests were performed for selecting the key esters of Langjiu. The panellists attended meetings twice a week for 1.5 hours each for three weeks. Triangular omission tests for key esters in Langjiu: only one compound was omitted (Table 4; tests 1 to 14) from the 14 esters, and then compared with the samples of all the key esters (14 esters). The ester concentration was the actual concentration of the ester in Honghualang (with an ethanol level of 53% (*v/v*)). In the triangulation test, each group had to randomly arrange three coded samples: one different sample and two identical samples. All of the liquor samples were shown to volunteers three times. The volunteer selected samples containing aroma compounds in three samples, although they were unsure. The results were based on published, tabulated data and were statistically analysed according to the binomial law of the distribution of answers in such tests.

#### 2.6.4. Determination of Odour Thresholds and OAVs

Through the omission experiment, the selected key esters were mixed in pairs, and the olfactory threshold of the binary mixture was measured in an aqueous solution of 53% ethanol and was conducted while using three alternative forced selection tests (3-AFC). The OAV was used to determine the contribution of aroma substances to the aroma of the liquor. The OAV was the ratio of the concentration of aroma substance to the threshold of the substance.

#### 2.6.5. Determination of Intensity of Binary Mixtures

Water solutions of 1-butanol were prepared at 25 ± 1 °C, according to the odor intensity referencing scale (OIRS, from level 1 (aqueous solution of 10 ppm) to level 12 (20,480 ppm)). The binary mixtures of ethyl acetate and ethyl hexanoate, and ethyl acetate and ethyl 2-methylbutyrate were mixed at the same strength, and the strength of the mixture was determined. The experiment was repeated three times.

### 2.7. Perceptual Interaction Analysis

#### 2.7.1. Feller’s Additive Model 

The olfactory threshold of mixed aroma substances was established. The results of all three alternative forced selection tests were statistically analysed. The results were summarised and presented as a detection probability and detection confidence of chemical stimulus. The detection probability was given by:(1)P = (m × p(c) − 1)/(m − 1),
where *P* = detection probability corrected for chance, *m* = number of choices per trial (this article, three), and *p*(c) = proportion correct (number of correct trials/total number of trials).

The sigmoid (logistic) equation was employed to model the psychometric function for groups and each individual, as follows:(2)$$P\;=\;\frac{1}{{1\;+\;e}^{-\frac{x\;-\;c}{D}}},$$
where *c* is olfactory threshold of the odorant (log µg/L), where x represents odorant concentration (log µg/L), where *P* is detection probability (0 ≤ *P* ≤ 1), and D is a parameter characteristic of each odorant that defines the gradient of the function [12,13,14].

Feller’s additive model could be used to evaluate the interactive effects of the mixtures [14]. The actual model that was obtained from the mixture experiment was compared to a simple additive theoretical model. The detection probability formula of the mixture *P*(AB) was as follows:(3)P(AB) = P(A) + P(B) − P(A)P(B),
where *P*(A) represents the probability of detecting component A and *P*(B) is the probability of detecting component B. If the sum of probabilities was higher than the panel’s detection performance for the mixture, some degree of suppression had occurred relative to statistical independence, in accordance with statistical independence, a certain degree of inhibition had occurred. On the contrary, some form of mutual addition or synergy had occurred. Furthermore, no mixing interaction occurred if the sum of probabilities matches was equal to the detection performance.

#### 2.7.2. The Odour Activity Value Approach

Ferreira V. [15] proposed that the odour activity values (OAVs) or concentration/threshold ratios of the odorant mixture at the threshold between the binary mixtures were approximately additive. That is, if a mixture contains *n* odorants and the sum of n concentrations/thresholds is y, then the mixture is above the threshold by y times. In arithmetic form:(4)$$\mathrm{OAVmix}=\sum_{j}^{n}\mathrm{OAV}i,$$
wherein the OAVmix refers to the number of times that the mixture was diluted to reach the threshold, and OAV*i* was the proportional concentration/threshold of the *i*th odorant of the mixture (the threshold was measured separately). OAVmix was originally defined as the threshold of the mixture and was recorded as Tm. Subsequently, as Ti and Ci were the thresholds and concentrations of the ith component of the mixture, respectively, individual OAVi values were calculated, added, and divided by the threshold of the mixture. This parameter was called *X*:(5)X = ∑CiTiTm = ∑OAViTm = ∑OAViOAVmix,
*X* values of 1 represent odour additivity, while a reduced value represents increased interaction or synergy. *X* values greater than 1 means that antagonism occurs [16].

#### 2.7.3. The Vector Model

The vector model could be thought of as an adjacent edge of a parallelogram, where the length of the edge represents the perceived intensity of the unmixed component, while the length of the diagonal in the figure represents the perceived intensity of the mixture [17]. Therefore, the OI of the binary mixture was successfully correlated with the odour intensity of its unmixed components, as follows:(6)$${\mathrm{OI}}_{ab}^{2}{\;=\;\mathrm{OI}}_{a}^{2}{\;+\;\mathrm{OI}}_{b}^{2}{\;+\;2\;\times \cos\alpha }_{ab}{\;\times \;\mathrm{OI}}_{a}{\mathrm{OI}}_{b\;},$$
where *a* and *b* represent two different substances, and OI*_ab_* is the OI of a mixture of *a* and *b*. The interaction coefficient cosα (where α is the angle between the sides of the parallelogram) represents the degree of interaction between the two unmixed components of the binary odour mixture.

In general, different odour mixtures had different values of cosα, which were usually based on experience to determine the components with equal perceptual intensity and they were used to predict the OI of the remaining mixtures in a group. For special cases where the perceptual intensities of the two odour components were equal, Equation (6) can be rewritten, i.e., OI*_a_* = OI*_b_*, and the following equation applies [18]:(7)$${\mathrm{OI}}_{ab}\;=\;\left({\mathrm{OI}}_{a}{\;+\;\mathrm{OI}}_{b}\right)\cos\frac{1}{2}\alpha .$$

The vector value (OIab) can be used to replace the actual aroma intensity of the mixture since the vector model is a perfect predictor of the aroma intensity of the mixture. 

### 2.8. Statistical Analysis

Analysis of variance (ANOVA) analysed the concentration of volatile compounds, and the interaction of esters in the Feller’s additive model and the vector model was analysed by Sigma Plot 12.0 (SYSTAT) software (Systat Software Inc, San Jose, CA, USA). The level of statistical significance was 5% (*p* < 0.05).

## 3. Results and Discussion

### 3.1. Qualitative and Quantitative Analysis

The qualitative and quantitative analysis of esters in langjiu was carried out to more accurately reveal the perceptual interaction between esters in Langjiu. Through GC-O sniffing and identification analysis, 31 ester compounds were found in the three kinds of Langjiu, application of GC-O to the liquors revealed 17, 17, and 16 aroma compounds (FD ≥ 16) in HHL, ZPL, and LPLJ, respectively (Table 1). The differences in the number of aroma compounds (FD ≥ 16) were mainly caused by concentration differences. These aroma compounds were determined by comparison with authentic standards, retention indices, and aroma descriptions. HHL contains more aroma substances, among which ethyl hexanoate (1024), ethyl 3-methyl butyrate (256), ethyl butyrate (256), ethyl 2-methylpropionate (128–256), ethyl 2-methylbutanoate (128–256), ethyl valerate (128–256), and ethyl caprylate (128–512) have higher dilution factors in three kinds of Langjiu. Ethyl ester compounds were important contributors to the pleasant fruit and floral aroma of Chinese liquor, according to reports in the literature [19]. These esters were mainly formed by the metabolism of yeast, filamentous fungi, etc., or fatty acid esterification reaction during fermentation [20].

Table 2 shows the concentrations and relative deviations of these compounds in Langjiu. Among these esters, ethyl acetate (450,892–529,294 µg/L) was the most abundant, followed by ethyl lactate (340,025–428,330 µg/L); in addition, ethyl propionate (32,654–35,598 µg/L), ethyl butyrate (23,585–27,387 µg/L), ethyl hexanoate (6078–13,849 µg/L), and ethyl 3-methyl butyrate (11,615–12,795 µg/L) were also present in higher concentrations. Wei, L. and Zhang, L. [5] used dichloromethane as the extractant for determining the main aroma components of Langjiu by gas chromatography-mass spectrometry (GC-MS). A total of 31 trace components were identified, and the most abundant were: ethyl hexanoate, hexanoate acid, ethyl lactate, acetic acid, and butyric acid. This was slightly different from the research results of Wei and others. This might have been due to the different extraction methods used to isolate aroma substances.

### 3.2. Determination of Key Compounds

#### 3.2.1. Threshold and OAV of Ester Compounds in Langjiu

Although GC-O analysis was an effective means of aroma compound identification, it did not accurately indicate the contribution of aroma compounds to the overall aroma. In liquor samples, aroma substances at a concentration above the detection threshold also contribute to the overall aroma. Therefore, individual OAVs were calculated to assess the contribution of different aromatic compounds to the aroma [21].

The aroma activity values of 24 ester aroma compounds in Langjiu were calculated by referring to the smell threshold of aroma substances in alcohol solution in the literature, and based on the quantitative results in different kinds of Langjiu. Table 3 shows the OAV calculation showed that the aroma contribution of each compound. It was found that 16 kinds of ester compounds have a greater contribution to the aroma of Langjiu (OAV ≥ 1), among which 16 kinds of HHL, 15 kinds of ZPL, and 15 kinds of LPLJ. Among the ester compounds, ethyl 3-methyl butyrate (OAV: 1801–1857), ethyl valerate (OAV: 333–389), ethyl butyrate (OAV: 289–336), ethyl caprylate (OAV: 212–228), ethyl isobutyrate (OAV: 168–179), and ethyl hexanoate (OAV: 109–250) have the highest OAV values among the three types of Langjiu. These esters were considered as key aroma substances in the studies of Gao et al. [8], and these esters were also key aroma substances in maotai-flavour liquor [22].

#### 3.2.2. Omission Analysis

HHL had most kinds of esters and the content of various esters therein was relatively high, according to Table 1, Table 2 and Table 3. Subsequently, taking the actual effect of the content into account, HHL was selected for subsequent testing and analysis. From Table 1 and Table 3, it can be seen that there are 14 esters with FD ≥16 and OAV ≥1. These 14 esters are important aroma compounds, namely ethyl acetate, ethyl propionate, ethyl 2-methylpropionate, ethyl butyrate, ethyl 2-methylbutyrate, ethyl 3-methyl butyrate, isoamyl acetate, ethyl valerate, ethyl hexanoate, ethyl caprylate, ethyl caprate, ethyl phenylacetate, ethyl 3-phenylpropionate, and ethyl lactate. During the experiment, the concentration of each compound was mixed with the actual concentration of HHL. Afterwards, triangular omission tests were carried out: only one compound was omitted (Table 4; tests 1 to 14) among 14 esters, so that a sample containing all of the studies compounds (14 esters) was compared with that only omitting one compound. For each group in triangulation tests, three coded samples were randomly arranged: one different and two identical. Through the omission testing of each compound, the results showed that these compounds had a significant effect on the overall aroma of the ester mixture. For ethyl acetate, ethyl 2-methylbutyrate, ethyl 3-methyl butyrate, ethyl hexanoate, and ethyl phenylacetate, the results showed that the difference was significant with *p* < 0.001. This was inconsistent with the conclusions of Fan et al. [25], because the liquor used and the pre-treatment methods were inconsistent.

#### 3.2.3. Selection of Five Ester Aroma Compounds

The results showed that esters made a significant contribution to the overall aroma of liquor [22]. Furthermore, through the study of the ester compounds in three kinds of Langjiu, GC-MS and GC-O technology identified 31 ester compounds. The key aroma components were further screened by omission test (*p* < 0.001). Five key esters were selected, respectively, ethyl acetate (*p* < 0.001), ethyl 2-methylbutyrate (*p* < 0.001), ethyl 3-methyl butyrate (*p* < 0.001), ethyl hexanoate (*p* < 0.001), and ethyl phenylacetate (*p* < 0.001). Finally, the perceptual interaction between the five esters was studied by using Feller’s additive model, odour activity values, and a vector model.

### 3.3. Olfactory Properties of Compounds

It is unreasonable to consider the overall aroma of Langjiu as the sum of the flavour contributions of each compound, because the interaction of different senses affecting flavour perception will be ignored, although the threshold of aroma compounds can be used as an indicator of their influence on flavour. Therefore, the interaction between substances was studied through the change of threshold before and after mixing.

#### 3.3.1. Studying the Olfactory Properties of Compounds by Feller’s Additive Model 

The change of threshold between the binary mixtures of key esters was revealed, and the experimental results were analysed, to investigate the interaction between the binary mixtures. The interaction between aromatics was studied by Feller’s additive model.

The five key ester compounds screened by omission experiment were mixed according to the proportion of their actual concentration in HHL. A total of ten groups of compounds were used to study the interaction of key ester compounds, namely: acetate and ethyl 2-methylbutyrate mixed, ethyl acetate and ethyl 3-methyl butyrate mixed, ethyl acetate and ethyl hexanoate mixed, ethyl acetate and ethyl phenylacetate mixed, ethyl 2-methylbutyrate and ethyl 3-methyl butyrate mixed, ethyl 2-methylbutyrate and ethyl hexanoate mixed, ethyl 2-methylbutyrate and ethyl phenylacetate mixed, ethyl 3-methyl butyrate and ethyl hexanoate mixed, ethyl 3-methyl butyrate and ethyl phenylacetate mixed, and ethyl hexanoate and ethyl phenylacetate mixed. The probability of detection of the binary mixture could be calculated by Feller’s additive model, and then estimated by the Feller model threshold, as derived from the sigmoid (logistic) equation.

The detection probabilities calculated using Feller’s addition model were lower than the actual detection probabilities that were obtained by the experiment (Figure 1), when ethyl acetate and ethyl 2-methylbutyrate were mixed (Figure 1a) (*p* = 0.057). An additive interaction occurred. A synergistic interaction occurred when: ethyl acetate and ethyl 3-methyl butyrate were mixed (Figure 1b) (*p* = 0.024), ethyl acetate and ethyl hexanoate were mixed (Figure 1c) (*p* = 0.011), ethyl acetate and ethyl phenylacetate were mixed (Figure 1d) (*p* = 0.005), ethyl 2-methylbutyrate and ethyl 3-methyl butyrate were mixed (Figure 1e) (*p* = 0.020), ethyl 2-methylbutyrate and ethyl hexanoate were mixed (Figure 1f) (*p* = 0.016), ethyl 2-methylbutyrate and ethyl phenylacetate were mixed (Figure 1g) (*p* = 0.002), ethyl 3-methyl butyrate and ethyl hexanoate were mixed (Figure 1h) (*p* = 0.016), ethyl 3-methyl butyrate and ethyl phenylacetate were mixed (Figure 1i) (*p* = 0.016), and when ethyl hexanoate and ethyl phenylacetate were mixed (Figure 1j) (*p* = 0.014). Cometto-Muñiz et al. [27] investigated the olfactory detectability of ethyl propionate and ethyl heptanoate, and measured the concentration detection (i.e., psychometric) function of individual and mixture odours at different concentrations. The results showed that the mixture approaches the response-addition model at low detection levels, i.e., the independence of the assay, while they approach the dose-addition model at a high detection level.

The ratio of the actual detection threshold obtained by the experiment and the theoretical threshold calculated by the Feller additive model, the lowest ratio of ethyl 2-methylbutyrate and ethyl phenylacetate was 0.10 (Figure 1g), and the highest ratio of ethyl acetate and ethyl 2-methylbutyrate was 0.57, according to the experimental results (Figure 1a). It could be seen from Figure 1 that the interaction between different aromatic compounds was different, which might be due to various factors such as the molecular size of the aromatic compounds themselves, the types of functional groups and their own volatility, as well as different intermolecular van der Waals forces and hydrogen bonds [28].

#### 3.3.2. Studying the Olfactory Properties of Compounds by the OAV Approach

The OAV has been applied to a large number of binary, ternary, and more complex mixtures, so it could be used to confirm the interaction between key ester compounds in Langjiu. Binary mixture OAVmix and ∑OAVi were calculated while using Equation (4), and the difference between the two was compared using Equation (5). Table 5 shows the experimental results. The ten groups of mixtures all have *X* < 1, the ethyl acetate and ethyl 2-methylbutyrate mixed was an additive effect, and the other groups shows a synergistic, which was consistent with results from Feller’s additive model. Many researchers have conducted extensive research into the OAV approach: Guadagni et al. [29,30] studied compounds containing nitrogen and sulphur in potatoes and found that these compounds have different effects on the overall aroma of the potato. Laura et al. [31] studied nine important oxidation-related aldehydes while using the OAV approach, revealing the interaction (addition or synergy) with other volatile compounds in wine. For example, the ratio of OAVmix to ∑OAVi of a mixture of (E)-2-hexenal, (E)-2-octenal, and (E)-2-nonenal was 3, which showed a synergistic effect. In addition, the (E)-2-enoyls were negatively correlated with the flavour of red wine, while branched aldehydes could enhance the dryness of the fruit and mask the negative effects of (E)-2-alkenals on the flavour of red wine.

#### 3.3.3. A Vector Model of Perceptual Odour Interaction

A binary mixture of ethyl acetate and ethyl hexanoate and a binary mixture of ethyl acetate and ethyl 2-methylbutanoate were selected by vector model since the vector model can distinguish the interaction between the two mixtures. Yan et al. [17] used a vector model to study the relationship between binary mixtures of aldehydes and ester binary mixtures, and the results evinced good correlation. Ethyl acetate is similar in structure to ethyl hexanoate. Five groups of equal interaction ethyl acetate and ethyl hexanoate were used for binary mixing and the strength of the mixture was determined. $\cos\frac{1}{2}\alpha$ was calculated according to Equation (7), and the binary substances were then obtained. The interactive relationship (Figure 2) is such that $\cos\frac{1}{2}\alpha \;=0.8072$. The ethyl acetate and ethyl 2-methylbutanoate with different structures were selected for analysis. The result revealed that $\cos\frac{1}{2}\alpha$ = 0.6577. The vector model can directly study the interaction between aroma substances, which was helpful in finding the law of interaction between aroma substances.

## 4. Conclusions

Qualitative and quantitative analyses of volatile esters in three kinds of Langjiu by GC-O and GC-MS with headspace solid phase microextraction (HS-SPME) were undertaken. A total of 31 ester compounds were identified, and 31 of them were quantitative analysis. FD value (FD ≥ 16), OAV (OAV ≥ 1), and omission test screened the key esters, and the results showed that ethyl acetate, ethyl 2-methylbutyrate, ethyl 3-methyl butyrate, ethyl hexanoate, and ethyl phenylacetate contributed to the aroma of Langjiu to a significant extent. Through the study of the interaction of binary mixtures in key esters by Feller’s additive model, OAV, and a vector model, it was confirmed that these ester compounds had additive or synergistic effects. Trace aroma components in liquor, especially the esters, have great influence on the flavour and quality of liquor, which is one of the important bases to judge the quality of liquor. The experimental results provide a scientific basis for the analysis and determination of liquor flavour substances and the evaluation of sensory quality, and they are of guiding significance for the improvement of liquor fermentation technology to improve the aroma quality of liquor.

## Figures and Tables

**Figure 1 foods-09-00183-f001:**
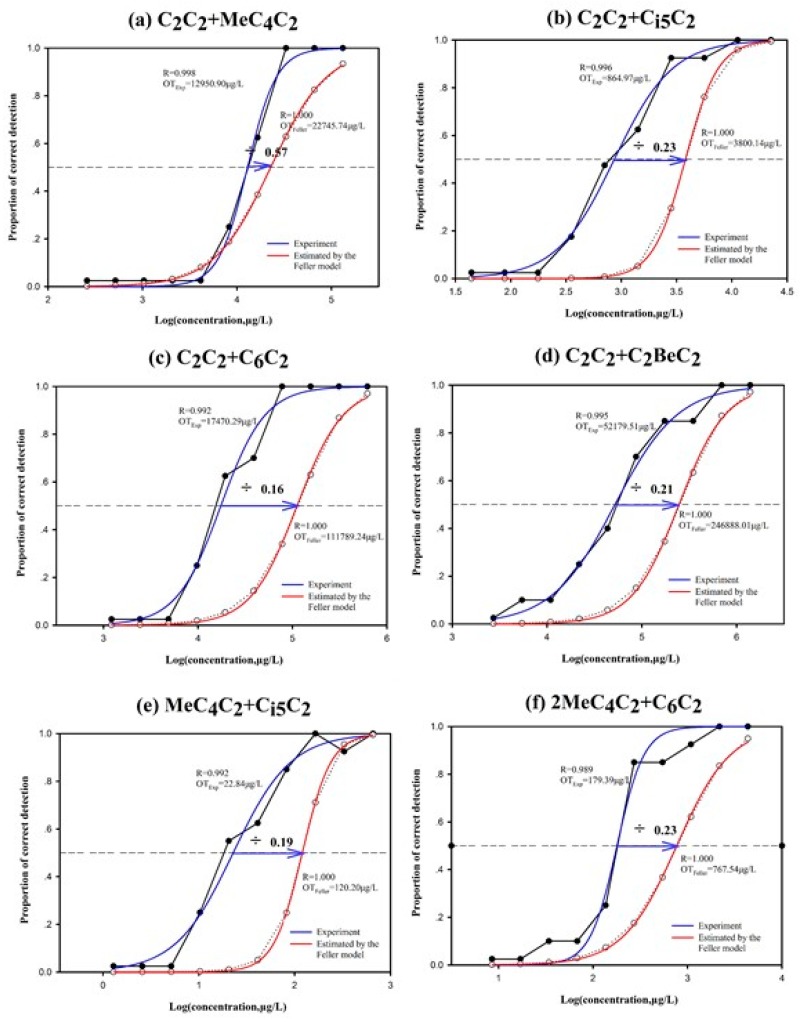
Effect of ethyl acetate and ethyl 2-methylbutyrate mixed (**a**), ethyl acetate and Ethyl 3-methyl butyrate mixed (**b**), ethyl acetate and ethyl hexanoate mixed (**c**), ethyl acetate and ethyl phenylacetate mixed (**d**), ethyl 2-methylbutyrate and Ethyl 3-methyl butyrate mixed (**e**), ethyl 2-methylbutyrate and ethyl hexanoate mixed (**f**), ethyl 2-methylbutyrate and ethyl phenylacetate mixed (**g**), Ethyl 3-methyl butyrate and ethyl hexanoate mixed (**h**), Ethyl 3-methyl butyrate and ethyl phenylacetate mixed (**i**), ethyl hexanoate and ethyl phenylacetate mixed (**j**). OT, olfactory threshold. The curves are drawn according to a sigmoid function. 2MeC_4_C_2_+C_i5_C_2_, ethyl 2-methylbutyrate and ethyl 3-methyl butyrate mixed.

**Figure 2 foods-09-00183-f002:**
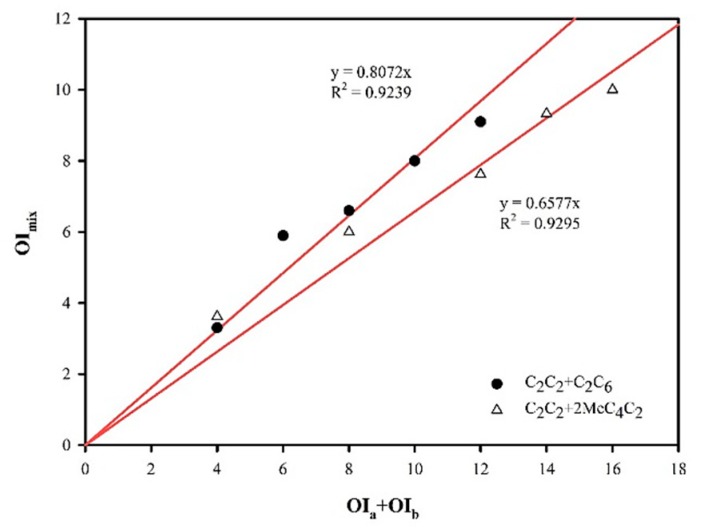
The relationship between OImix of a binary odor mixture (OIa+OIb) and the summation of its unmixed constituents’ odor intensities (OIa+OIb).

**Table 1 foods-09-00183-t001:** Aroma compounds identified by gas chromatography-olfactometry (GC-O) in three Langjiu.

No.	RI		Aroma Compound	Descriptor	Basis of ID ^a^	FD ^c^		
	HP-Wax	DB-5				HHL ^d^	ZPL ^e^	LPLJ ^f^
Ester								
1	897	638	Ethyl acetate	Pineapple	Aroma,RI,MS,S	64	512	64
2	967	726	Ethyl propionate	Banana	Aroma,RI,MS,S	16	32	16
3	974	773	Ethyl 2-methylpropionate	Sweet, Rubber	Aroma,RI,MS,S	128	256	256
4	983		Propyl acetate	Fruity	Aroma,RI,MS,S	1	1	1
5	1020	788	Isobutyl acetate	Fruity,Banana	Aroma,RI,MS,S	2	1	1
6	1045	815	Ethyl butyrate	Apple	Aroma,RI,MS,S	256	256	256
7	1059	863	Ethyl 2-methylbutyrate	Apple	Aroma,RI,MS,S	128	256	256
8	1074	868	Ethyl 3-methyl butyrate	Fruity	Aroma,RI,MS,S	256	256	256
9	1127	890	Isoamyl acetate	Banana	Aroma,RI,MS,S	16	16	16
10	1141	914	Ethyl valerate	Apple	Aroma,RI,MS,S	128	256	256
11	1222		Butyl butyrate	Banana, Pineapple	Aroma,RI,MS,S	8	4	4
12	1246	1017	Ethyl hexanoate	Ppple peel fruit	Aroma,RI,MS,S	1024	1024	1024
13	1278	1072	Isoamyl butyrate	Green apple	Aroma,RI,MS,S	1	1	1
14	1324	1109	Caproic acid propyl ester	Pineapple	Aroma,RI,MS,S	1	1	1
15	1342	1113	Ethyl heptanoate	Fruity	Aroma,RI,MS,S	4	4	4
16	1350	830	Ethyl lactate	Green fruity	Aroma,RI,MS,S	16	16	16
17	1443	1213	Ethyl caprylate	Fruity, Fat	Aroma,RI,MS,S	512	128	128
18	1465	1268	Isopentyl hexanoate	Pineapple	Aroma,RI,MS,S	1	1	1
19	1542		Ethyl nonanoate	Fruity	Aroma,RI,MS,S	4	4	4
20	1616	1404	Hexyl hexanoate	Vegetable fruity	Aroma,RI,MS,S	1	nd ^b^	1
21	1634	1069	Ethyl 2-furoate	Floral,burnt	Aroma,RI,MS,S	2	2	2
22		1113	Ethyl caprate	Fruity	Aroma,RI,MS,S	16	8	8
23	1684	1189	Ethyl benzoate	Floral	Aroma,RI,MS,S	8	8	8
24	1801	1263	Ethyl phenylacetate	Rosy, Honey	Aroma,RI,MS,S	32	32	32
25	1851	1617	Ethyl laurate	Waxy, Floral	Aroma,RI,MS,S	16	16	16
26	1904	1369	Ethyl 3-phenylpropionate	Fruity,floral,wine	Aroma,RI,MS,S	32	32	16
27	2056	1782	Ethyl tetradecanoate	Floral	Aroma,RI,MS,S	1	1	nd ^b^
28	2109	1884	Ethyl pentadecanoate	Honey sweet	Aroma,RI,MS,S	4	4	4
29	2265	2023	Palmitic acid ethyl ester	Fruity, Creamy	Aroma,RI,MS,S	16	16	16
30	2468	2200	Ethyl oleate	Fatty	Aroma,RI,MS,S	16	16	16
31	2512	2194	Linoleic acid ethyl ester	Fatty	Aroma,RI,MS,S	8	16	1

^a^ Aroma compounds were identified by comparison to reference standards by GC-O; RI, compounds were identified on HP-Wax and DB-5 by comparison of reference standard. S, compounds were identified by authentified standards. ^b^ nd, not detected. ^c^ FD, flavour dilution factor. ^d^ HHL, Honghualang. ^e^ ZPL, Zhenpinlang. ^f^ LPLJ, Langpailangjiu. ^g^ GC-O: gas chromatography-olfactometry.

**Table 2 foods-09-00183-t002:** Standard curves and concentrations of 31 odorants in three type Langjiu by SPME-GC-MS.

No.	Aroma Compound	Quantitative Ion (m/z)	Standard Curve Slope	Intercept	*R* ^2^	Basis of ID	HHL ^c^		ZPL ^d^		LPLJ ^e^	
							av (µg/L)	RSD (%) ^a^	av (µg/L)	RSD (%)	av(µg/L)	RSD (%)
Ester												
1	Ethyl acetate	43	0.062	0.0475	0.991	MS,RI,Std	488,275	6	529,294	4	450,892	7
2	Ethyl propionate	57	0.215	−0.0517	0.996	MS,RI,Std	32,654	3	35,598	3	35,567	5
3	Ethyl 2-methylpropionate	43	0.406	−0.0682	0.992	MS,RI,Std	9803	5	10,305	3	9664	9
4	Propyl acetate	43	0.259	−0.0042	0.996	MS,RI,Std	3831	8	4625	4	3263	6
5	Isobutyl acetate	43	0.665	−0.0061	0.996	MS,RI,Std	541	4	576	6	514	6
6	Ethyl butyrate	71	0.427	−0.1793	0.995	MS,RI,Std	23,585	3	25,975	7	27,387	5
7	Ethyl 2-methylbutyrate	57	1.168	−0.1376	0.993	MS,RI,Std	3961	3	3944	9	3792	5
8	Ethyl 3-methyl butyrate	88	0.6	−0.1378	0.995	MS,RI,Std	12,795	6	12,412	6	11,615	5
9	Isoamyl acetate	43	1.169	−0.0989	0.996	MS,RI,Std	2872	4	2892	5	2745	2
10	Ethyl valerate	88	0.886	−0.1134	0.994	MS,RI,Std	8922	5	9817	5	10,438	2
11	Butyl butyrate	71	2.857	−0.0373	0.991	MS,RI,Std	207	5	184	4	189	9
12	Ethyl hexanoate	88	1.43	−0.6541	1.000	MS,RI,Std	13,849	4	6078	3	9868	3
13	Isoamyl butyrate	71	4.816	−0.0715	0.993	MS,RI,Std	293	6	262	5	267	5
14	Propyl hexanoate	99	3.705	−0.0736	0.996	MS,RI,Std	775	7	548	2	545	5
15	Ethyl heptanoate	88	3.801	−0.406	1.000	MS,RI,Std	6404	1	4215	6	4378	6
16	Ethyl lactate	45	0.016	0.0133	0.994	MS,RI,Std	340,025	6	414,676	8	428,330	5
17	Ethyl caprylate	88	5.104	−0.1023	0.993	MS,RI,Std	2733	6	2938	2	2801	3
18	Isopentyl hexanoate	70	11.64	−0.099	0.999	MS,RI,Std	454	7	202	6	188	7
19	Ethyl nonanoate	88	9.773	−0.1131	0.999	MS,RI,Std	174	5	197	3	155	5
20	Hexyl hexanoate	117	9.222	−0.0604	0.998	MS,RI,Std	287	4	nd ^b^		123	9
21	Ethyl 2-furoate	95	0.695	0.0004	0.999	MS,RI,Std	1608	6	2013	3	1429	4
22	Ethyl caprate	88	12.08	−0.2581	0.999	MS,RI,Std	1311	7	336	7	318	6
23	Ethyl benzoate	105	5.3	−0.0488	0.998	MS,RI,Std	320	5	316	5	360	3
24	Ethyl phenylacetate	91	5.723	−0.016	0.993	MS,RI,Std	1699	4	2007	4	1842	6
25	Ethyl laurate	88	8.637	−0.3412	1.000	MS,RI,Std	898	4	598	3	567	4
26	Ethyl 3-phenylpropanoate	104	6.304	−0.065	0.998	MS,RI,Std	418	4	457	6	482	4
27	Ethyl tetradecanoate	88	2.621	−0.0099	1.000	MS,RI,Std	699	5	568	5	nd ^b^	
28	Ethyl pentadecanoate	88	1.274	−0.024	0.998	MS,RI,Std	543	4	1135	4	562	5
29	Palmitic acid ethyl ester	88	0.467	0.1301	0.998	MS,RI,Std	17,021	6	17,391	4	2507	7
30	Ethyl oleate	55	0.743	0.0004	0.997	MS,RI,Std	606	8	530	2	141	3
31	Linoleic acid ethyl ester	67	0.123	−0.0003	0.996	MS,RI,Std	2521	4	40,743	7	222	5

^a^ RSD, relative standard deviation. ^b^ nd, not detected. ^c^ HHL, Honghualang. ^d^ ZPL, Zhenpinlang. ^e^ LPLJ, Langpailangjiu.

**Table 3 foods-09-00183-t003:** Odour activity value (OAV) of the volatile compound in Langjiu.

No.	Compound	Odor Threshold (µg/L)	OAV		
			HHL	ZPL	LPLJ
1	Ethyl acetate	32,600 ^a^	15	16	14
2	Ethyl propionate	19,000 ^a^	2	2	2
3	Ethyl 2-methylpropionate	57.5 ^a^	170	179	168
4	Propyl acetate	4740 ^e^	<1	<1	<1
5	Isobutyl acetate	922 ^a^	<1	<1	<1
6	Ethyl butyrate	81.5 ^b^	289	319	336
7	Ethyl 2-methylbutyrate	18 ^d^	220	219	210
8	Ethyl 3-methyl butyrate	6.89 ^a^	1857	1801	1686
9	Isoamyl acetate	93.93 ^a^	31	31	29
10	Ethyl valerate	26.8 ^a^	333	366	389
11	Butyl butyrate	110 ^e^	2	2	2
12	Ethyl Hexanoate	55.3 ^a^	250	109	178
13	Isoamyl butyrate	20 ^e^	15	13	13
14	Caproic acid propyl ester	12,783.77 ^b^	<1	<1	<1
15	Ethyl heptanoate	13,200 ^a^	<1	<1	<1
16	Ethyl lactate	128,000 ^a^	3	3	3
17	Ethyl caprylate	12.9 ^a^	212	228	217
18	Isopentyl hexanoate	1400 ^d^	<1	<1	<1
19	Ethyl nonanoate	3150 ^a^	<1	<1	<1
20	Ethyl caprate	1120 ^a^	1	<1	<1
21	Ethyl benzoate	1430 ^a^	<1	<1	<1
22	Ethyl phenylacetate	407 ^a^	4	5	5
23	Ethyl laurate	1500 ^c^	<1	<1	<1
24	Ethyl 3-phenylpropionate	125 ^a^	3	4	4

^a,b^ Odor thresholds were determined in 46% ethanol/water solution and they were taken from Gao et al. (2014) [8], Wenlai Fan et al. (2011) [23], µg/L. ^c^ Odor threshold taken from Jiang Bao et al.(2013) [24], µg/L; ^d^ Odor threshold taken from Fan Haiyan, Fan Wenlai & Xu, Yan (2015) [25], µg/L; ^e^ Odor threshold taken from Gemert L.J.V. (2003) [26], µg/L.

**Table 4 foods-09-00183-t004:** Olfactory impact of the omission of various esters from complex aromatic reconstitutions.

	C_2_C_2_	C_3_C_2_	C_i4_C_2_	C_4_C_2_	MeC_4_C_2_	C_i5_C_2_	C_2_C_i5_	C_5_C_2_	C_6_C_2_	C_8_C_2_	C_7_C_2_	C_2_BeC_2_	PrBeC_2_	2OHC_3_C_2_	Difference Observed
Complete TAR in HHL	x	x	x	x	x	x	x	x	x	x	x	x	x	x	
Test 1	-	x	x	x	x	x	x	x	x	x	x	x	x	x	***
Test 2	x	-	x	x	x	x	x	x	x	x	x	x	x	x	**
Test 3	x	x	-	x	x	x	x	x	x	x	x	x	x	x	*
Test 4	x	x	x	-	x	x	x	x	x	x	x	x	x	x	**
Test 5	x	x	x	x	-	x	x	x	x	x	x	x	x	x	***
Test 6	x	x	x	x	x	-	x	x	x	x	x	x	x	x	***
Test 7	x	x	x	x	x	x	-	x	x	x	x	x	x	x	*
Test 8	x	x	x	x	x	x	x	-	x	x	x	x	x	x	**
Test 9	x	x	x	x	x	x	x	x	-	x	x	x	x	x	***
Test 10	x	x	x	x	x	x	x	x	x	-	x	x	x	x	**
Test 11	x	x	x	x	x	x	x	x	x	x	-	x	x	x	=
Test 12	x	x	x	x	x	x	x	x	x	x	x	-	x	x	***
Test 13	x	x	x	x	x	x	x	x	x	x	x	x	-	x	**
Test 14	x	x	x	x	x	x	x	x	x	x	x	x	x	-	**

***, 0.1% significant level; **, 1% significant level; *, 5% significant level; = no significant difference; x, presence of listed compounds; and -, absence of listed compounds. C_2_C_2_, Ethyl acetate; C_3_C_2_, Ethyl propionate; C_i4_C_2_, Ethyl 2-methylpropionate; C_4_C_2_, Ethyl butyrate; 2MeC_4_C_2_, Ethyl 2-methylbutyrate; C_i5_C_2_, Ethyl 3-methyl butyrate; C_2_C_i5_, Isoamyl acetate; C_5_C_2_, Ethyl valerate; C_6_C_2_, Ethyl hexanoate; C_8_C_2_, Ethyl caprylate; C_7_C_2_, Ethyl caprate; C_2_BeC_2_, Ethyl phenylacetate; PrBeC_2_, Ethyl 3-phenylpropionate; 2OHC_3_C_2_, Ethyl lactate.

**Table 5 foods-09-00183-t005:** The OAV approach to study the perceptual interaction between esters.

No.	Mixture	OAVmix	OAV1	OAV2	∑OAVi	*X*	Interaction
1	C_2_C_2_+2MeC_4_C_2_	38.01	1.32	21.01	22.33	0.59	additive effect
2	C_2_C_2_+C_i5_C_2_	579.29	1.32	131.75	133.06	0.23	synergistic effect
3	C_2_C_2_+C_6_C_2_	28.74	1.32	3.53	4.84	0.17	synergistic effect
4	C_2_C_2_+C_2_BeC_2_	9.39	1.32	0.79	2.11	0.22	synergistic effect
5	2MeC_4_C_2_+C_i5_C_2_	733.62	21.01	131.75	152.75	0.21	synergistic effect
6	2MeC_4_C_2_+C_6_C_2_	99.28	21.01	3.53	24.53	0.25	synergistic effect
7	2MeC_4_C_2_+C_2_BeC_2_	204.40	21.01	0.79	21.80	0.11	synergistic effect
8	Ci_5_C_2_+C_6_C_2_	388.68	131.75	3.53	135.27	0.35	synergistic effect
9	C_i5_C_2_+C_2_BeC_2_	548.80	131.75	0.79	132.54	0.24	synergistic effect
10	C_6_C_2_+C_2_BeC_2_	15.58	3.53	0.79	4.32	0.28	synergistic effect

C_2_C_2_+2MeC_4_C_2_, ethyl acetate and ethyl 2-methylbutyrate mixed; C_2_C_2_+C_i5_C_2_, ethyl acetate and ethyl 3-methyl butyrate mixed; C_2_C_2_+C_6_C_2_, ethyl acetate and ethyl hexanoate mixed; C_2_C_2_+C_2_BeC_2_, ethyl acetate, and ethyl phenylacetate mixed; 2MeC_4_C_2_+C_i5_C_2_, ethyl 2-methylbutyrate, and ethyl 3-methyl butyrate mixed; 2MeC_4_C_2_+C_6_C_2_, ethyl 2-methylbutyrate and ethyl hexanoate mixed; 2MeC_4_C_2_+C_2_BeC_2_, ethyl 2-methylbutyrate, and ethyl phenylacetate mixed; C_i5_C_2_+C_6_C_2_, ethyl 3-methyl butyrate, and ethyl hexanoate mixed; C_i5_C_2_+C_2_BeC_2_, ethyl 3-methyl butyrate, and ethyl phenylacetate mixed; C_6_C_2_+C_2_BeC_2_, ethyl hexanoate, and ethyl phenylacetate mixed.

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
