# Peer review of "Evaluation of Perceptual Interactions between Ester Aroma Components in Langjiu by GC-MS, GC-O, Sensory Analysis, and Vector Model"

_foods, 2020, doi:10.3390/foods9020183_

Round 1

Reviewer 1 Report

The idea that the individual OAVs were calculated to assess the contribution of different aromatic compounds to the aroma is in my opinion very good point

Noteworthy are the large number of compounds analyzed and how they are interpreted.

Statistical modelling and mathematical formulas make it easier to read and organize the results. However, authors should emphasize more clearly the innovation of research. In its current form it is not clear

Author Response

Thank you very much for these valuable comments. The questions and unclear statements have been interpreted in this study, and your questions have been answered. We sincerely hope to get your guidance and reply.

Response to Reviewer 1 Comments

General Comments:

The idea that the individual OAVs were calculated to assess the contribution of different aromatic compounds to the aroma is in my opinion very good point

Noteworthy are the large number of compounds analyzed and how they are interpreted.

Statistical modelling and mathematical formulas make it easier to read and organize the results. However, authors should emphasize more clearly the innovation of research. In its current form it is not clear

 Point 1: The idea that the individual OAVs were calculated to assess the contribution of different aromatic compounds to the aroma is in my opinion very good point

Noteworthy are the large number of compounds analyzed and how they are interpreted.

Response 1: Thank you for your suggestion. (1) In this paper, the esters in Lang wine were analyzed qualitatively and quantitatively by GC-MS, GC-O. These aroma compounds were determined by comparison with authentic standards, retention indices, and aroma descriptions, as shown in P6, Line260-262. (2) The key esters in Langjiu were selected by AEDA (FD≥16), OAV (OAV≥1), omission experiment (P<0.1%), and the interaction between esters was studied by Feller's additive model, OAV, and a vector model.

Point 2: Statistical modelling and mathematical formulas make it easier to read and organize the results. However, authors should emphasize more clearly the innovation of research. In its current form it is not clear.

Response 2: Thank you for your suggestion. The innovation of this paper is (1). The characteristic aroma components in Langjiu were identified by the method of AEDA, OAV, and omission test. (2) The synergistic effect of esters was studied by Feller's additive model, OAV, and a vector model, as shown in P2, Line 62-70.

Reviewer 2 Report

Your paper:" Evaluation of perceptual interactions between ester aroma components in Langjiu by GC-MS, GC-O, sensory analysis and vector model" is scientifically interesting. It shows the application of different analytical techniques coupled by statistical analysis  for the characterization and enhancement of a typical Chinese liquor. In order to be able to extrapolate details that could preserve it from counterfeits. 

Unfortunately, I didn't found in the Conclusions the answers to all the listed obiectives such as the identification of marker aromatic compounds of Langjiu liquor and the guidelines to improve the quality. 

Then, I believe that is necessary to increase and strengthen the results with the missing informations.

Typos are highlighted in the text

Author Response

Thank you very much for these valuable comments. The questions and unclear statements have been interpreted in this study, and your questions have been answered. We sincerely hope to get your guidance and reply.

Response to Reviewer 2 Comments

General Comments:

Your paper:" Evaluation of perceptual interactions between ester aroma components in Langjiu by GC-MS, GC-O, sensory analysis and vector model" is scientifically interesting. It shows the application of different analytical techniques coupled by statistical analysis for the characterization and enhancement of a typical Chinese liquor. In order to be able to extrapolate details that could preserve it from counterfeits.

Unfortunately, I didn't found in the Conclusions the answers to all the listed obiectives such as the identification of marker aromatic compounds of Langjiu liquor and the guidelines to improve the quality.

Then, I believe that is necessary to increase and strengthen the results with the missing informations.

Typos are highlighted in the text

Point 1: Unfortunately, I didn't found in the Conclusions the answers to all the listed obiectives such as the identification of marker aromatic compounds of Langjiu liquor and the guidelines to improve the quality.

Response 1: Thank you for your suggestion. The marker aromatic compounds of Langjiu is ethyl acetate, ethyl propionate, ethyl 2-methylpropionate, ethyl butyrate, ethyl 2-methylbutyrate, ethyl 3-methyl butyrate, isoamyl acetate, ethyl valerate, ethyl hexanoate, ethyl caprylate, ethyl caprate, ethyl phenylacetate, ethyl 3-phenylpropionate, and ethyl lactate, as shown in P12, Line 323-327. Through the omission experiment, five key esters were selected, namely ethyl acetate, ethyl 2-methylbutyrate, ethyl 3-methyl butyrate, ethyl hexanoate, and ethyl phenylacetate, as shown in P12, Line 328-338. In this paper, the interaction between key esters was studied, which can guide the liquor brewing technology and improve liquor quality.

Round 2

Reviewer 2 Report

I believe the manuscript can be accepted.

Author Response

Thank the reviewers for their approval of the article. In addition, I have read the whole article and corrected the grammatical errors. I hope the article can be accepted after correction.
